# Novel Crosslinking System for Poly-Chloroprene Rubber to Enable Recyclability and Introduce Self-Healing

**DOI:** 10.3390/polym13193347

**Published:** 2021-09-29

**Authors:** Anureet Kaur, Julien E. Gautrot, Gabriele Cavalli, Douglas Watson, Alan Bickley, Keizo Akutagawa, James J. C. Busfield

**Affiliations:** 1School of Engineering and Materials Science, Queen Mary University of London, London E1 4NS, UK; Anureet.kaur@qmul.ac.uk (A.K.); j.gautrot@qmul.ac.uk (J.E.G.); g.cavalli@qmul.ac.uk (G.C.); akutag@jcom.home.ne.jp (K.A.); 2Weir Advanced Research Centre, Glasgow G1 1RD, UK; douglas.watson@mail.weir (D.W.); alan.bickley@mail.weir (A.B.)

**Keywords:** polychloroprene, thiol-ene, disulfide metathesis, self-healing, recycling, rubber

## Abstract

The introduction of dynamic bonds capable of mediating self-healing in a fully cross-linked polychloroprene network can only occur if the reversible moieties are carried by the cross-linker itself or within the main polymer backbone. Conventional cross-linking is not suitable for such a purpose. In the present work, a method to develop a self-healable and recyclable polychloroprene rubber is presented. Dynamic disulfide bonds are introduced as part of the structure of a crosslinker (liquid polysulfide polymer, Thiokol LP3) coupled to the polymer backbone via thermally initiated thiol-ene reaction. The curing and kinetic parameters were determined by isothermal differential scanning calorimetry and by moving die rheometer analysis; tensile testing was carried to compare the tensile strength of cured compound, healed compounds and recycled compounds, while chemical analysis was conducted by surface X-ray Photoelectron Spectroscopy. Three formulations with increasing concentrations of Thiokol LP-3 were studied (2, 4, 6 phr), reaching a maximum ultimate tensile strength of 22.4 MPa and ultimate tensile strain of 16.2 with 2 phr of Thiokol LP-3, 11.7 MPa and 10.7 strain with 4 phr and 5.6 MPa and 7.3 strain with 6 phr. The best healing efficiencies were obtained after 24 h of healing at 80 °C, increasing with the concentration of Thiokol LP-3, reaching maximum values of 4.5% 4.4% 13.4% with 2 phr, 4 phr and 6 phr, respectively, while the highest recycling efficiency was obtained with 4 phr of Thiokol LP-3, reaching 11.2%.

## 1. Introduction

Polychloroprene, also commonly known as Chloroprene Rubber (CR) or Neoprene, was first commercialized in 1931 by DuPont [1]. It was one of the very first synthetic rubbers and it is still one of the most widely used elastomers because of its versatile properties in a wide range of applications. Depending on the specific grade of CR used, cross-linking can occur using a metal oxides-based method, such as the combination of zinc oxide and magnesium oxide, or by using a combined system of sulfur and metal oxides [2]. 

Compared to other synthetic rubbers, such as butadiene rubber (BR), styrene-butadiene rubber (SBR), butyl rubber (IIR), ethylene propylene diene monomer rubber (EPDM) and nitrile rubber (NBR), conventionally cross-linked CR has excellent abrasion resistance, good ageing and weather resistance and excellent oxidation and ozone resistance properties, secondary only to hydrogenated nitrile rubber (HNBR) [3]. Vulcanized CR articles are widely used for hoses and gaskets and, in particular, in the minerals extraction industry because they are resistant to extreme conditions including erosion and abrasion. In some applications, such as the inner lining of pumps and hoses for slurry transportation, 40% of the initial mass can be lost due to the erosion activity performed by the slurry whilst the remaining 60% is disposed of at the end of its life as waste. For many applications, the conditions of temperature and flow rate generate a rapid erosion rate of the liners which requires frequent replacement of the inner lining. We anticipate that the introduction of a fast intrinsic self-healing mechanism might help to reduce the high erosion rate in these types of application, reducing the number of interventions required to replace the liner over the pump’s life, which will save money both through the pump being in service for longer and by reducing the number of refits required. In addition, and reflecting on the first principle of Green Chemistry, which states that “It is better to prevent waste formation than to treat it after it is formed” [4], in order to reach zero waste after a refit, potential recycling approaches for CR are also now required. To enable recyclability and introduce self-healing, a novel method using thiol-ene radical coupling on CR with a liquid polysulfide polymer carrying dynamic disulfide bonds is reported here for the first time. 

The chemical structure of CR gives the rubber its very useful properties. These make it suitable for applications where a resistance to extreme environments including highly oxidative chemistry, organic solvents, oils and high temperatures is required [3]. This is possible thanks to the vinylic position of the chlorine atom in the repeating trans-1,4 units for the majority of the double bonds in the network, as it inactivates the double bond by creating a strong dipole through the inductive effect, and thus making it less keen to participate in oxidation reactions. However, to justify the cross-linking process, previous researchers have proposed a reaction mechanism involving a small amount of defects in the raw material in the form of 1,2-adducts formed during emulsion polymerization of CR [5]. These are believed to undergo isomerism which, under specific conditions, transposes the chlorine atom to a less sterically hindered position as shown in Figure 1, a necessary rearrangement in order to have a reactive substrate [6].

Recyclability and self-healing behaviors are two highly desirable properties that will require both the curing-process and the final application properties to be considered in combination. In order to have a fully recyclable CR, one strategy is to develop a route for cross-linking the rubber that allows the links to be broken and reformed. In order to introduce an intrinsic self-healing mechanism, the cross-linking system has to be able to form dynamically to counter damage developed in service. A recent, exhaustive review by Behera et al. [7] reveals that most studies proposed different dynamic covalent and physical interactions for the development of self-healable elastomers. Given the low reactivity of uncured CR, there are not many examples of recyclable and/or self-healable elastomers based on CR in the literature. To the best of our knowledge, only one report of such materials was made by Xiang et al. [8], where a self-healing and recyclable material was developed by incorporating an organic complex of copper (II) methacrylate as a catalyst to trigger disulfide metathesis at 120 °C in a conventional type of cross-linking formulation where both metal oxides and sulfur systems were used. 

Disulfide metathesis-based intrinsic self-healing mechanisms have been widely implemented for diene-containing elastomers thanks to their ability to break and reform through a wide range of stimuli, such as chemical triggers including both redox [9] or alkaline environments [10], or they can be physically induced, for example by controlling their temperature [11] or through exposure to light [12]. It has also been demonstrated that the dynamic cleavage and reforming of the labile disulfide bond can occur spontaneously [13]. Nevejans et al. demonstrated that in absence of chemical triggers the radical-mediated [2 + 1] mechanism, as shown in Figure 2, is followed during the cleavage of the disulfide bond and its subsequent reforming [14].

The amount of disulfide bonds present in the commercially available sulfur modified CRs are insufficient to develop an efficient self-healing mechanism, as the emulsion polymerization process to synthetize these CR grades typically restricts sulfur addition to 0.6 wt%. Trying to compensate with a conventional curing system such as a sulfur based system in combination with a metal oxide system would create a cross-linked network where it would be difficult to quantify the amount of disulfide bonds in comparison to the amount of polysulfide bonds. In this work, thiol-ene radical coupling, a more specific and well-established reaction mechanism, widely applied in several polymer systems, was explored for the crosslinking of CR and conferring self-healing properties for the first time [15,16,17,18,19]. A thiol-ene-based approach was selected as it would be beneficial because the reaction parameters and conditions are mild and compatible with CR chemistry and processing, but would also allow compatibility with other crosslinking mechanisms. The use of liquid polysulfide polymer as an alternative cross-linker, introduced via a photo-initiated thiol-ene reaction on polybutadiene, was first investigated by Xiang et al. [20] where disulfide metathesis was triggered by UV light. More recently, Gao et al. [21] studied the introduction of different grades of liquid polysulfide polymers in acrylate-based elastomers via thermo-, photo- and redox-initiated thiol-ene reactions, where disulfide metathesis was triggered by heat and catalytic activity. Similarly, thiolated hydrophilic polymer backbones can be photo-crosslinked, photo-cleaved and self-healed via photo-oxydation and photo-reduction reactions [22].

In the present work, a new cross-linking method for CR is introduced, based on thiol-ene chemistry, in which a liquid polysulfide polymer, Thiokol LP-3, is introduced via a thermally initiated free radical reaction, according to the mechanism shown in Figure 3. A methodology to investigate the kinetic parameters of the thiol-ene curing reaction applied to CR is developed considering aspects such as the average molecular weight change in the mixing stage, heat evolution and torque values in isothermal conditions, while chemical characterization through different spectroscopy techniques, thermal properties characterization and tensile testing are used to establish a method to evaluate the self-healing behavior and the recyclability of the cured compounds examined here. 

## 2. Materials and Methods

### 2.1. Materials

Chloroprene Rubber (CR; SN 122 grade) was supplied by The Weir Group PLC (Glasgow, UK). Thiokol (TLP; LP-3 grade; mercaptan terminated liquid polysulfide polymer of n diethyleneoxymethane units connected with Sx linkages, where 6 ≤ N ≤ 42 and x predominantly equal to 2) was purchased from Wessington Group Ltd. (Reading, UK). 2,2′-Azobis(2-methylpropionitrile) (AIBN; 98% purity; free radical initiator), Stearic Acid (StA; 95% purity, reagent grade) and Chloroform-d (CDCl3; 99.8 atom % D) were purchased from Sigma-Aldrich Co Ltd. (Dorset, UK). Magnesium Oxide (MgO; ≥97% purity, Light Technical grade) and Zinc Oxide (ZnO; ≥98% purity, Light Technical grade) were purchased from VWR International Ltd. (Leicestershire, UK). Toluene (≥99.9% purity, HPLC grade) and Tetrahydrofuran (THF; ≥99.9% purity, inhibitor free, HPLC grade) were purchased from Honeywell International Inc. (Bracknell, UK).

In the present work CR/TLP blends were tested in different conditions, before and after curing, healing and recycling. The terminology adopted to refer to the blends in specific conditions is as follows:Compound: something that has been mixed, but not cured.Cured compound: new cured compound.Healed compound: cured compound healed after damage induction.Recycled compound: recycled cured compound.

### 2.2. Preparation of CR/TLP Blends

CR/TLP blend compositions are reported in Table 1. CR chips were masticated in a laboratory two-roll mill until a homogenous mixture was obtained (5–10 min). StA was incorporated first (10–15 min) to minimize roll sticking, reduce the mixture viscosity and hence improve the mixing process. Subsequently, MgO was gradually added in order to neutralize any traces of released hydrochloric acid during processing and vulcanization [2] (20–30 min). Once a homogenous mixture was obtained, TLP was gradually added (30 min for CR/TLP 2%, 50 min for CR/TLP 4%, 2 h for CR/TLP 6%). Finally, AIBN was added and the rubber was molded in a manual hot press at 170 °C, applying a constant pressure of about 12 MPa, for a curing time equal to *t*_90_ taken form the cure rheology, which was different for each mixture, as reported in Table 2.

### 2.3. Recycling of CR/TLP Blends

Cured compounds were masticated in the two roll mill until a coarse granulate was obtained, as shown in Figure 1. The material was then molded in the manual hot press at 170 °C for the same time as for the related cured compound.

### 2.4. Characterization

#### 2.4.1. Average Molecular Weight

The number average molecular weight (*M*_n_), the weight average molecular weight (*M*_w_) and the polydispersity index (PD) of CR, TLP and each compound were determined by performing Gel Permeation Chromatography (GPC) using Agilent Technologies 1260 Infinity GPC/SEC system equipped with a refractive index (RI) detector. Calibration was carried out using polystyrene standards (Mw range from 162 g/mol to 6,570,000 g/mol) from Agilent Technologies, Inc. Samples were prepared at least 48 h prior to the analysis by dissolving approximately 10 mg of sample in 5 mL of THF. The solutions were then filtered through a 0.2 μm PTFE syringe filter and transferred into GPC vials. Each run was performed by injecting 100 μL in the columns, kept at a constant temperature of 25 °C, with a flow rate of fresh THF of 1 mL/min for 40 min.

#### 2.4.2. Isothermal and Dynamic Differential Scanning Calorimetry (DSC) 

Isothermal DSC was performed on each compound to investigate the curing behavior while dynamic DSC was conducted to determine the glass transition temperature of all cured compounds and all recycled compounds. Both analyses were carried out using TA Instruments DSC25 equipment. All samples were placed in Tzero Aluminium Hermetic pans, and the normalized heat flow was measured for all conditions. The curing exotherms for each compound were measured for 1 h at 160 °C, 170 °C and 180 °C. The thermal behavior of the cured compounds and recycled compounds was studied by conducting heat–cool–heat experiments as follows: (1) first heating ramp from −90 °C to 200 °C at the rate of 10 °C/min; (2) second cool ramp from 200 °C to −90 °C at the rate of 5 °C/min; (3) third heat ramp from −90 °C to 200 °C at the rate of 10 °C/min. The *T*_g_ was calculated using the half height–midpoint type method.

#### 2.4.3. Curing Behavior

The curing behavior for each compound was determined using a Monsanto Moving Die Rheometer (MDR) 2000, with the lower die moving at 1.66 Hz. Each analysis was performed at temperatures of 160 °C, 170 °C and 180 °C for 3 h. The degree of cure was determined for each temperature using the following Equation (1):(1)$$\begin{array}{c} \alpha =\frac{S_{t}-S_{0}}{S_{t100}-S_{0}} \end{array}$$
where *α* is the degree of cure, *S*_t_ is the torque at curing time t, *S*_0_ is the torque value at time 0 and *S*_t100_ is the torque value at the end of the curing reaction. From the isothermal DSC analysis the degree of cure was calculated using the following Equation (2):(2)$$\begin{array}{c} \alpha =\frac{∆H_{t}}{∆H_{\mathrm{tot}}} \end{array}$$
where ∆*H*_t_ is the heat evolved up to time t and ∆*H*_tot_ is the total heat evolved during the curing reaction, calculated by integrating the heat flow curve in the isothermal DSC thermogram. The rate of the reaction was determined from the differential of the degree of cure curve. The differential equation of *α* as a function of *t* was plotted versus *α* and a non-linear fitting to the autocatalytic model (3) following the Levenberg–Marquardt method was performed:(3)$$\begin{array}{c} \frac{d\alpha }{dt}=k\left(T\right)\cdot \alpha^{m}{\left(1-\alpha \right)}^{n} \end{array}$$
where m and n are parameters determining the reaction order for an autocatalytic reaction and *k*(*T*) is the Arrhenius function determined by the following expression (4):(4)$$k\left(T\right)=k_{0}\cdot e^{-\frac{E_{a}}{RT}}$$
where *k*_0_ is the pre-exponential factor, *E*_a_ is the activation energy, *T* is the curing temperature expressed in Kelvin and **R** is the gas constant equal to 8.31446 J mol^−1^ K^−1^. The obtained values of *k*(*T*) for each temperature were plotted in the form of ln[*k*(*T*)] versus 1/*T* and, after linear regression, *E*_a_ and *k*_0_ were determined from the slope and the intercept values, respectively.

#### 2.4.4. Cross-Link Density

Equilibrium swelling experiments were carried out in toluene and Flory–Huggins theory was applied to calculate the cross-link density of cured compounds and recycled compounds using the following Equations (5)–(7):(5)ln(1−VE)+VE+ξVE2+2ρEν0[X]phys(VE)13=0
(6)$$\begin{array}{c} V_{E}=\frac{V_{\mathrm{EN}}}{V_{\mathrm{EN}}+V_{S}} \end{array}$$
(7)$$\begin{array}{c} V_{\mathrm{EN}}=\frac{m_{\mathrm{EN}}}{m_{\mathrm{tot}}}\cdot \frac{m_{0}}{\rho_{E}} \end{array}$$

After measuring the mass of the samples, *m*, these were left in toluene at room temperature for 14 days in absence of light. The swollen samples were then removed from the solution, tap dried quickly to remove any excess liquid and transferred in a calibrated container to be weighed again to measure *m*_0+S_. The samples were then dried under reduced pressure using a rotatory evaporator, and the weight of the dried unswollen sample, *m*_0_, was measured; *ρ*_E_ is the density of the sample; *m*_EN_ is the mass of the rubber and cross-linker used in the whole formulation, whose total mass is *m*_tot_. *V*_EN_, the volume of elastomer network, can be determined with Equation (6) and *V*_E_, the volume fraction of elastomer, can be calculated according to Equation (5), where *V*_S_ is the volume of the solvent absorbed. In Equation (4) *ξ* is the Flory–Huggins interaction parameter, which is 0.14 for chloroprene rubber–toluene interaction [23], *ν*_0_ is the molar volume of the solvent, the volume occupied by one mole of solvent, which is 106.28 cm^3^ for toluene, and [*X*]_phys_ is the total number of cross-links, given by chemical and physical interactions.

#### 2.4.5. Raman Spectroscopy

Raman spectroscopy was performed using a Renishaw inVia confocal Raman microscope. Each compound was exposed to a laser wavelength of 785 nm for 60 s while cured compounds and recycled compounds were exposed to the laser for 30 s only. Spectra were collected over the range of 400–1800 cm^−1^.

#### 2.4.6. Chemical Analysis

X-ray photoelectron spectroscopy (XPS) analysis of CR, cured compounds and recycled compounds was carried out using the Thermo Scientific Nexsa XPS System. The analysis method consisted of using a laser spot size of 100 µm for each acquisition, and each sample was subject to ion beam etching for 30 s, using ion energy of 2000 eV in cluster mode. For each sample a survey scan was performed in the binding energy range of −10 to 1350 eV, with a step size of 1 eV, acquiring 10 scans with a dwell time of 10 ms. Subsequently, a high resolution element spectra for sulfur, oxygen, magnesium, carbon and chlorine was recorded with a 0.1 eV energy step size and pass energy of 50 eV, by performing 10 scans with a dwell time of 50 ms. The survey spectra interpretation was performed using Avantage software. Multipeak analyzing method was used to perform deconvolution of the high resolution spectra and peak identification was conducted by comparing the experimental values to the literature [24].

### 2.5. Mechanical Characterization

#### 2.5.1. Tensile Testing

Tensile testing until failure was carried on cured compounds, healed compounds and recycled compounds. Dumbbell shape samples were cut out using an ISO-37-4 die cutter 24 h after curing was completed. Testing was carried out using an Instron 5967 machine equipped with a 1 kN load cell and pneumatic side action grips, using a rate of 1 mm/s. The width and the thickness were measured prior to the start of the test while the initial length, *L*_0_, was measured after gripping the sample, ensuring the width of the sample remained constant in the gauge length [25]. Uniaxial stress and strain were calculated using the following Equations (8) and (9):(8)$$\begin{array}{c} \sigma_{\mathrm{eng}}=\frac{F}{A_{0}} \end{array}$$
(9)$$\begin{array}{c} \varepsilon =\frac{∆L}{L_{0}} \end{array}$$
where *σ*_eng_ is the engineering stress, *F* is the measured force, *A*_0_ is the initial unstrained cross-sectional area, *ε* is the strain and ∆*L* the measured elongation. Tests on cured compounds were repeated until failure for a total of 30 times for each CR/TLP blend formulation. The failed samples were then exposed to several different healing conditions after this initial testing each of which was repeated at least five times. The testing of the recycled samples were also repeated at least 5 times.

#### 2.5.2. Healing Conditions

Healing was evaluated for two different types of damage on the dumbbell samples, the first being the healing ability of the sample after an initial break during a tensile test and the second after severing the sample with a razor in a direction perpendicular to the length (that is, a transverse through-cut). After the damage was introduced, all the samples were quickly placed back in contact and left either at room temperature or 80 °C. Different samples were allowed to recover for 1 h, 24 h or 48 h. Tensile tests were performed on all the samples again until failure, and 5 samples were tested for each different healing condition.

#### 2.5.3. Healing Efficiency and Recycling Efficiency

Healing efficiency, *η*_h_, was calculated by comparing the ultimate tensile strength after healing to the average value of ultimate tensile strength of the cured compound using the following Equation (10):(10)$$\begin{array}{c} \eta_{h}\%=\frac{\sigma_{\mathrm{healed}}}{average\;\sigma_{\mathrm{cured}\;\mathrm{compound}}}\cdot 100 \end{array}$$

The recycling efficiency, *η*_r_, was calculated in the same way using the following Equation (11) as it can be considered to be a type of healing:(11)$$\begin{array}{c} \eta_{r}\%=\frac{\sigma_{\mathrm{recycled}}}{average\;\sigma_{\mathrm{cured}\;\mathrm{compound}}}\cdot 100 \end{array}$$

## 3. Results and Discussion

The mixing stage on the two roll mill has two effects on the compound when it is being prepared. First, the mastication process reduces the average starting molecular weight, and second, it generates a rise in temperature which promotes the initial chemical interactions between the formulation ingredients. As disulfide metathesis is favored at elevated temperatures, and both CR and TLP contain in their main structure disulfide bonds, the exchange reaction described in Figure 4a is inevitable. Appendix A shows the shift in retention times for each formulation for the signals related to CR chains and TLP chains, as both increase as the concentration of TLP is increased. The calculated *M*_n_ and *M*_w_ values reported in Table 3 show how the mixing time required for blending CR and TLP can impact the final molecular weight distribution, resulting in higher mastication, thus smaller chains. This reflects the lower molecular weights as longer times are required to obtain a homogenous mixture as the quantity of TLP is increased. The PD values, however, indicate a more homogeneous distribution of *M*_w_ for CR/TLP 2% and CR/TLP 4% compared to CR, while CR/TLP 6% shows a higher polydispersity value. This can be related to the disulfide metathesis occurring during the mixing stage as the blending of TLP into the CR matrix is the result of the dilution-type effect carried by the amount of disulfide bonds introduced with TLP, equivalent to the amount of disulfide bonds already present in CR if 2% of TLP is used. For the CR/TLP 2% formulation, the mixing stage almost reached an equilibrium between the metathesis effect and the mastication process. When the amount of disulfide bonds introduced is doubled, as for CR/TLP 4%, or tripled, as for CR/TLP 6%, the mastication process is more pronounced, the PD values are higher, and the equilibrium with the metathesis effect takes longer. Appendix A reports the Raman spectra comparisons between each compound, TLP and CR, and a clear difference between CR and TLP is noted, considering the multipeak region of the –C–S– signals of 580–700 cm^−1^, indicating the presence of chemically different –C–S– units (thiols, disulfide and polysulfide related), and –S–S– signals at 510 cm^−1^ [26]. Raman spectra of the cured compounds are reported in Appendix A. Appendix A does not show significant quantitative results, as the samples induce laser scattering, but the presence and preservation of disulfide and polysulfide bonds is confirmed in both cured compounds and recycled compounds.

The isothermal DSC and the rheology analysis both suggest that multiple reactions take place during cross-linking. In Figure 2a, the curve related to the isothermal DSC of CR/TLP 2% carried at 160 °C clearly shows three peaks, each one related to a specific chemical interaction, and the time difference between the interactions reduces as the temperature increases. Appendix A, related to CR/TLP 4% and CR/TLP 6%, respectively, show that by increasing the concentration of TLP, the occurrence of these different chemical events is more condensed. The calculation of the total heat evolved during the analysis and the subsequent calculation of the degree of cure is shown for CR/TLP 2% in Figure 2b, while Appendix A report the degree of cure and total heat evolved for CR/TLP 4% and CR/TLP 6%, respectively. The amount of heat evolved increases with the temperature, except for CR/TLP 4%, but the highest heat evolution is obtained for CR/TLP 2%. The fitting of the autocatalytic model (Figure 2c for CR/TLP 2%, Appendix A for CR/TLP 4%, and Appendix A for CR/TLP 6%) was performed by dividing each curve into two or three domains, depending on the distinctive peaks shown. CR/TLP 2% shows three domains in the 160 °C and 170 °C analyses, whereas the 180 °C analysis only shows two domains, suggesting that two events are now taking place simultaneously. CR/TLP 4% and CR/TLP 6% only show two domains in the 160 °C and 180 °C analysis, while the 170 °C analysis can be fitted in three domains for both compounds. The calculated *E*_a_ values from Figure 1d for CR/TLP 2%, and from Appendix A for CR/TLP 4% and CR/TLP 6%, respectively, are reported in Table 4. *E*_a_ for Reaction 1 (R1) decreases with increasing concentration of TLP, while it increases for Reaction 2 (R2) and Reaction 3 (R3). R1 and R2 are not really following the linear Arrhenius Equation (4) as is R3; in fact, a curved Arrhenius plot is generated, which is more pronounced in R2. The negative values of *E*_a_ for R2 are an indication of multiple reactions occurring at the same time, and the overall rate of the reaction decreases with increasing temperature [27]. This could be the domain where the thiol-ene free-radical reaction is taking place between 2 to 4 min, where *α* values are between 0.2 and 0.4.

The kinetic and curing parameters’ calculations from the MDR are shown in Figure 3 for CR/TLP 2%, and in Appendix A for CR/TLP 4% and CR/TLP 6%, respectively. Each shows a similar trend to the data obtained from isothermal DSC; however, the derived *E*_a_ values for R1, R2 and R3, reported in Table 5, are all positive, and they only appear to increase as the concentration of TLP increases. For CR/TLP 6%, only one fitting per temperature was performed as no significant peak was detectable. The different kinetic and curing results obtained from the DSC and MDR analyses indicate that they cannot be compared for the characterization of the thiol-ene curing reaction. The isothermal DSC allows the calculation of kinetic values that are related to heat evolution changes that depend on the type of chemical interaction occurring, whereas the kinetic parameters derived from the MDR analysis are related only to the cross-linked network changes.

Dynamic DSC analyses, reported in Figure 4a, show that the *T*_g_ values reported in Table 6 are shifting in the cured compounds to slightly higher temperatures compared to the *T*_g_ of CR, and even higher *T*_g_ values are obtained from recycled compounds. The temperature of crystallization, *T*_c_, is higher in the cured compounds compared to CR, and it is even higher in recycled compounds, but it decreases with increasing concentrations of TLP. 

The cross-link density calculations, reported in Table 7, highlight an increase in recycled compounds, with increasing concentration of TLP, as shown in Figure 4b. In CR/TLP 2%, the amount of cross-linking points is probably too low, so during recycling only weak chemical interactions are restored and not enough entanglements are generated to compensate the amount of lost permanent covalent bonds.

The chemical degradation in the transition from cured compound to recycled compound was assessed with XPS analysis, as shown in Figure 5. In particular, an attempt was made to quantify the chemical species in each cured compound and recycled compound. The deconvolution details can be found in the Appendix A for O 1s spectra, in Appendix A for Cl 2p spectra, in Appendix A for Mg 1s spectra, in Appendix A for S 2p spectra and in Appendix A for C 1s spectra. The calculated area of each deconvolution is reported in Appendix A for O 1s spectra, in Appendix A for Cl 2p spectra, in Appendix A for Mg 1s spectra, in Appendix A for S 2p spectra and in Appendix A for C 1s spectra.

In CR the total atomic weight of oxygen is 12.2%, 27% is detected as inorganic oxides as talc (formula Mg_3_Si_4_O_10_(OH)_2_) which is present in the raw material, and the remaining 73% is detected as organic-oxide, as some CR could probably be oxidized during storage. The atomic weight of oxygen is reduced in cured compounds (11.9% for CR/TLP 2%, 7.4% for CR/TLP 4%, 10.8% for CR/TLP 6%) while it increases in recycled compounds (13.8% for R-CR/TLP 2%, 12.6% for R-CR/TLP 4%, 12.5% for R-CR/TLP 6%). The amount of organic oxygen increases with the concentration of TLP in the cured compounds, in agreement with the chemical structure of TLP, but it increases even more in the recycled compounds, suggesting that auto-oxidation is taking place. The total atomic weight of chlorine in CR is 6.6%, of which 7% is inorganic and 93% is organic. It remains comparable in the cured compounds (7.1% for CR/TLP 2%, 11.7% for CR/TLP 4%, 5.1% for CR/TLP 6%). The recycled compounds show a similar trend, but the detected total composition is reduced (none for R-CR/TLP 2%, 3.2% for R-CR/TLP 4%, 2.8% for R-CR/TLP 6%). The magnesium composition is only detectable in the survey spectra of recycled compounds (1.8% for R-CR/TLP 2%, 1.4% for R-CR/TLP 4%, 1.6% for R-CR/TLP 6%). Magnesium is detected as an oxide and as a chloride salt, which is formed when MgO reacts with HCl during degradation. The concentration of MgCl_2_ appears to increase at the highest concentration of TLP tested. The total amount of sulfur in CR is 1.7%, and in the high resolution spectrum this is detected as 35% organic sulfur with the following breakdown likely: 13% in polysulfide bonds and 12% in disulfide bonds. The rest of it is detected in inorganic form, likely as: 59% sulfite (SO_3_^2−^) and 17% sulfate (SO_4_^2−^). In the cured compounds (0.7% for CR/TLP 2%, 1.3% for CR/TLP 4%, 1.0% for CR/TLP 6%), sulfur is detected in lower amounts. The disulfide peak shows an increase as the concentration of TLP is increased, and a new peak for CR/TLP 6% is detectable at 165.8 eV, accounting for 3% of the total amount, which could be related to thiol, –SH, species. This peak increases for the recycled compounds (total amount of sulfur: 0% for R-CR/TLP 2%, 1.1% for R-CR/TLP 4%, 1.0% for R-CR/TLP 6%), but the quantity of disulfide and polysulfide bonds is reduced while the overall quantity of inorganic sulfur increases. The total atomic weight of carbon in CR is 79.4%. The amounts of specific species are calculated as: 56% -C-**C**-C-, 46% -C-**C**-O- and 2% -C-**C**-Cl. The amount of -C-**C**-O- increases from cured compounds to recycled compounds, while -C-**C**-Cl decreases. CR/TLP 4% and R-CR/TLP 4% does not follow the trend.

Tensile testing of cured compounds show the initial cured behavior is promising. CR/TLP 2% has an average maximum tensile strain of 16.2 and an average maximum tensile strength of 22.4 MPa; CR/TLP 4% showed an average maximum tensile strain of 10.7 and an average maximum tensile strength of 11.7 MPa; while CR/TLP 6% showed an average maximum tensile strain of 7.3 and an average maximum tensile strength of 5.6 MPa. There is a large reduction in UTS and fracture strain for all of the healed samples. Figure 6a–c for healed compounds (CR/TLP 2%, CR/TLP 4% and CR/TLP 6%, respectively) show that deliberately cut, healed compounds consistently recovered better than those that underwent a previous tensile test to failure. Figure 6d shows that higher healing efficiencies are obtained as the concentration of TLP, the healing time and healing temperatures increase. We found that 47% of the CR/TLP 2% that were healed after failure samples did not have any residual strength after healing. In addition, 33% of the results were obtained from samples healed at r. t., while only 20% of the healed at 80 °C could be tested. The highest healing efficiency was obtained after 24 h of healing at 80 °C (maximum 3.1%, average 1.8%). All of the CR/TLP 2% healed after cutting samples gave stress–strain curves and the highest healing efficiency was obtained after 24 h of healing at 80 °C (maximum 4.5%, average 3.9%). CR/TLP 4% shows a better behavior for the healed after failure samples as the percentage of not tested specimens was lower, at 23%, while 33% of the samples were tested after healing at r.t. and 44% after healing at 80 °C. The highest healing efficiency in this case was obtained after 48 h of healing at 80 °C (maximum 5%, average 2.3%) but the highest average value was obtained after 24 h at 80 °C (maximum 4.4%, average 3.0%). The healing after cutting of CR/TLP 4% did not give significant results for 10% of the samples, but only after healing at r.t., while all the samples healed at 80 °C performed well giving the highest healing efficiency after 24 h at 80 °C (maximum 6.5%, average 5.6%). We found that 93% of the CR/TLP 6% samples healed after failure did not heal, as only two samples were tested after healing at r.t. after 1 h and after 24 h and the healing efficiency value obtained was equal in both cases: 2.6%. The performance of healed samples after cutting was better, as only 23% of the samples, all healed at r.t., were not tested. The highest healing efficiency was obtained after 24 h at 80 °C (maximum 13.4%, average 12%).

Tensile testing of recycled compounds, reported in Figure 7a, only shows a partial recovery of the tensile properties. As reported in Figure 7b, the recycling efficiency increases as the concentration of TLP increases above 2%; however, the highest value is obtained for CR/TLP 4%. Unlike the other two recycled compounds, only 60% of the R-CR/TLP 2% samples could be tested to generate a stress–strain curve reaching the maximum value of recycling efficiency of 3%,with an average of 0.6%. R-CR/TLP 4% reached a maximum value of recycling efficiency of 11.2%, with an average of 9.8%, while R-CR/TLP 6% reached a maximum of 10.2%, with an average of 6.8%.

## 4. Conclusions

The compounds were analyzed with GPC and Raman techniques to evaluate the morphology of the mixtures prior to vulcanization, while isothermal DSC and MDR analysis were used to determine the kinetic and curing parameters. The tensile strength properties of the cured compound were compared to the ones of healed compounds and recycled compounds by calculating the healing and recycling efficiencies according to Equations (10) and (11), respectively. The comparison of the chemical and physical properties between the cured compounds and the recycled compounds was carried with Dynamic DSC, TGA, XPS analysis and cross-link density calculations. 

Isothermal DSC and MDR analysis cannot be compared directly to simultaneously evaluate the kinetic and curing parameters in a thiol-ene based cross-linking approach. From the isothermal DSC analysis, only R3, 0.4 < *α* < 1, in each compound follows the Arrhenius plot, and the *E*_a_ values increase with increasing concentration of TLP, suggesting that the domain of R3 could be related to the amount of sulfur in disulfide form. The domain of R2, on the other hand, is giving negative *E*_a_ values probably related to multiple reactions occurring simultaneously. Findings in the literature that explain this phenomenon are related to the presence of an intermediate complex which can exhibit both strongly curved Arrhenius plots and negative temperature dependence [27]. The MDR based Arrhenius plot calculations do not follow a linear relation, which makes it difficult to draw conclusions. It may be possible to apply some sort of time–temperature superposition such as the Williams–Landel–Ferry equation to determine the effect of rheological contributions [28], but this has not been attempted here. The reaction order parameters reported in the Appendix A (for isothermal DSC) and in Appendix A (for MDR), show that m varies from 0 to 1 and n from 1 to 8.04. This suggests that some complicated interactions are taking place. According to the chemical analyses conducted these could be: auto oxidation, disulfide metathesis, MgO mediated crosslinking and crystallization. At this stage the simple auto catalytic Equation (3) is not sufficient to represent these reactions.

The XPS data indicate that a detrimental oxidation processes resulting in MgCl_2_ formation increases with increasing concentrations of TLP in cured compounds, and the degradation is more pronounced in the recycled compounds. It also suggests that during recycling, many thiol groups are oxidized and do not contribute to network recovery, perhaps underlying the modest recovery of mechanical properties observed in tensile testing. It is worth mentioning that the XPS analysis reported in this work was conducted only on the fully cured surface of the sample. Further analysis on cut edges will have to be performed.

Better values of healing efficiency are obtained on deliberately cut samples, as opposed to tensile fractured samples. This suggests that the cross-linked network is destroyed during the breaking cycle and is not allowed to redevelop during the healing phase as the pre-existing stretch orients the chains in a way that impedes healing, though the details of this reorientation are not explored in great detail here. Yet, some evidence of this resides in the calculation of cross-link density for CR/TLP 2% and R-CR/TLP 2%, as we can consider the recycling process as a drastic healing event where disulfide bonds, entanglements and polar interaction rearrangements are taking place; compared to the 4% and 6% formulations, the 2% formulation does not show any increase in cross-link density when transitioning from cured compound to recycled compound. This is probably due to the high level of deformation the cured compound is subject to during recycling and how the rearrangement of the few disulfide and polysulfide bonds is not adequate to restore the previous mechanical properties, as the network is mainly relying on entanglements. 

The healed compound CR/TLP 6% shows the greatest healing properties compared to the other samples, but this is merely due to the fact that its baseline UTS is four times lower than the highest performing cured compound, CR/TLP 2%. In absolute terms, the healed compounds have similar values for UTS and failure strain.

In terms of the recycling efficiency, R-CR/TLP 4% performs best. This may be explained by the dynamic DSC analysis where CR/TLP 4% shows the highest value of *H*_cryst._. This could indicate that the presence of secondary interactions can improve healing and recycling properties in CR/TLP systems.

Overall, our data demonstrate the proof-of-concept of self-healing CR compounds using TLP and combinations of thiol-ene and disulfide metathesis reactions. Based on our results, it is apparent that the role of the dynamic (or labile) bonds that are necessary for the self-healing and recycling applications can be tracked using this combination of different characterization techniques. Using this approach allows these and potentially other curing systems to be examined and possibly optimized. Further research in these directions will help advance self-healing CR into a technology to a point where it should be suitable for commercial consideration. In particular, a more thorough understanding the degradation mechanism of thiol residues during recycling is expected to better retain mechanical properties post-healing. In addition, exploring other thiol/disulfide crosslinkers will help developing a rational design of tiol-ene crosslinked self-healable CRs.

## Data Availability

The data presented in this study are available on request from the corresponding author.

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
