# Peer review of "Novel Crosslinking System for Poly-Chloroprene Rubber to Enable Recyclability and Introduce Self-Healing"

_polymers, 2021, doi:10.3390/polym13193347_

Round 1
Reviewer 1 Report
The research paper entitled "Novel crosslinking system for poly-chloroprene rubber to enable recyclability and introduce self-healing" is an interesting work of polychloroprene rubber with some new findings.
The manuscript is well written and organized.
Some minor points should be addressed before publication:
- Abstract should add more quantitative information
- Introduction of polychloroprene should be in more detail. First and second paragraph should add references.
- Novelty should be established in Introduction.
- About characterization method, I suggest to use FT-IR to evaluate functional group of polymer. It is useful to compare FT-IR and Raman spectroscopy.
- The results of XPS are very not good. The curves are not smoothed.
- Results and Discussion should be combined because Discussion part is too less in section Discussion and conclusions.
Author Response
We thank the reviewer for their comments.
We are adding more quantitative information as suggested in the abstract (see changes in lines 23-27). We are also adding more details for polychloroprene and the relative references in the first two paragraphs (please see lines 37-41).
Regarding establishing the novelty in the introduction: please see lines 55-57.
Regarding the use of FT-IR: we characterized our materials via FTIR: please see the Figure in the attached file.
As we are mostly interested in S-S/S-H changes, the region of interest is usually between 450 and 500 cm-1 approximately, but usually these stretching bands are very weak and usually other absorptions overlap in this region, making difficult an accurate analysis, unless we drastically increase the concentration of Thiokol, which was not investigated in this study to avoid too much dilution of the main rubber backbone. We note that Raman spectroscopy would be more appropriate for such characterisation, as these S-S/S-H are more active in this type of spectroscopy, but Raman spectra of cured and recycled compounds shows a lot of scattering/ fluorescence, making necessary a change in sample preparation prior to performing spectroscopy.
Regarding XPS: the noise shown in the XPS graphs is in line with spectra published in the literature (including from the authors, in a range of peer-reviewed journals). This is particularly the case for weak peaks associated with atom present at low abundance in our samples. Smoothing is not typically performed, to avoid masking the real source data and introducing artefacts. Performing smoothing on Figure 5c, for example, would give highly inaccurate subsequent multi-peak fitting resulting in non-reliable quantification of the different chemical species detected. However, curve fitting is typically performed to extract different components from multiple overlapping peaks. This was indeed reported in our supplementary information.
We thank the reviewer for their suggestion to combine the results and discussions parts. As some of the discussion was already included in our results section, and our previous discussion section was indeed short, and rather aimed to provide conclusion comments, we renamed our results section as "Results and Discussion" and the discussion section as "Conclusions".

Reviewer 2 Report
In the manuscript entitled ‘Novel crosslinking system for poly-chloroprene rubber to enable recyclability and introduce self-healing’, the authors prepared cross-linked poly-chloroprene rubber with dynamic disulfide bonds. There are lots of interesting results. I have the following questions and my minor comments are listed below.
MINOR COMMENTS
- Page 15-16, there is no relative results discussion of the Figure 6(a), Figure 6(b), and Figure 6(d), please supply the corresponding description.
- Please further highlight the novelty of the article. In this work, the authors reported poly-chloroprene rubber with enable recyclability and introduce self-healing. The authors are encouraged to tell the main features and advantages of the poly-chloroprene rubber from other rubbers, and the following works (Nano-Micro Letters, 2021, 13: 91; Composites Communications, 2020, 22: 100486) may be for your description, to reveal the advantages or novelty of your work, finally to further highlight the theme.
- The authors mentioned that all the samples were quickly placed back in contact and left either at room temperature or 80°C. Why did you choose the room temperature or 80°C to get the self-healing rubber?
- The cited references can hardly reflect the main publications on which the work is based. Some important relative works should be reviewed or compared.
Author Response
We thank the reviewer for their comments.
Descriptions for Figure 6a, Figure 6b and Figure 6c have been added in our revised manuscript. The reference to Figure 6d was also corrected. Many thanks for spotting this mistake (please see lines 438-439).
https://link.springer.com/article/10.1007/s40820-021-00624-4
https://www.sciencedirect.com/science/article/pii/S245221392030156X
The above publications suggested have been consulted for the structure, and the novelty of our work has been more clearly detailed in our introduction (please see lines 37-41; 55-57).
Room temperature and 80°C were chosen for this work as this is the typical temperature range encountered with these materials in service.
Reviewer 3 Report
The entire manuscript is presented clearly and understandably. The article brings enough novelty to the subject matter to be published in this journal without difficulty. The language and style of the publication is very good. Presentation of results clear and easy to understand.
Good work.
Author Response
We thanks the reviewer for this positive comment.